# Estimating Patient Empowerment and Nurses’ Use of Digital Strategies: eSurvey Study

**DOI:** 10.3390/ijerph18189844

**Published:** 2021-09-18

**Authors:** Olga Navarro Martínez, Jorge Igual García, Vicente Traver Salcedo

**Affiliations:** 1Department of Nursing, Catholic University of Valencia, 46007 Valencia, Spain; 2Instituto ITACA, Universitat Politècnica de València, 46022 Valencia, Spain; vtraver@itaca.upv.es; 3Instituto de Telecomunicaciones y Aplicaciones Multimedia (ITEAM), Departamento de Comunicaciones, Universitat Politècnica de València, 46022 Valencia, Spain; jigual@dcom.upv.es

**Keywords:** health literacy, digital literacy, digital skills, patient empowerment

## Abstract

Patient empowerment is seen as the capability to understand health information and make decisions based on it. It is a competence that can improve self-care, adherence and overall health. The COVID-19 pandemic has increased the need for information and has also reduced the number of visits to health centers. Nurses have had to adapt in order to continue offering quality care in different environments such as the digital world, but this entails assessing the level of their patients’ empowerment and adapting material and educational messages to new realities. The aim of this study is, on the one hand, to assess nurses’ use of digital resources to provide reinforcing information to their patients and, on the other hand, to evaluate how they assess the level of empowerment of their patients. To perform the study, 850 nurses answered 21 questions related to their own digital literacy and patients’ empowerment. The ability to make decisions is the characteristic most selected by nurses (70%) as useful in measuring patient empowerment, whereas 9.19% do not measure it in any way. Printed material is most often used by nurses to offer additional information to patients (71.93%), mobile applications are the least used option (21.58%), and elder nurses are those who most recommend digital resources. In this study, younger nurses make little or no use of technology as a resource for training and monitoring patients. In spite of some limitations concerning the study, digital health needs to be promoted as an indisputable tool in the nurse’s briefcase in the future to ensure that older patients can manage electronic resources in different fields.

## 1. Introduction

One of the communication forms that has necessarily grown exponentially during the pandemic has been the digital. This paradigm change caused by the pandemic has affected in a very important way not only professionals who adapted new communication approaches but also patients, especially older patients with chronic pathology who, with some cause, we assume to have fewer digital skills and a lower health literacy.

The COVID-19 pandemic has led to important organizational and healthcare changes in hospitals and health centers to address the health emergency in the best possible way. In Spain, this has meant that follow-up programs for patients with chronic diseases have been halted, which has left part of the population unattended in terms of healthcare advice, mainly given and managed by nurses.

In a situation like the one we have experienced, the important need to hold patients responsible for their own self-care and promote their empowerment has become more apparent than ever.

Patient empowerment is a trend that aims to change the paternalistic healthcare system and empower patients, especially those with chronic conditions [1]. Patient empowerment is not exclusively a pandemic-related need but can be very useful in such cases. The WHO considers “empowerment” to be an essential concept of health promotion; indeed it is one of the key points of the European Strategy for the Prevention and Control of Noncommunicable Diseases, which states that people should be empowered to promote their own health, to interact with health services and to participate actively in disease management [2].

The conceptual model of patient empowerment that is discussed in this article refers to being informed, being able to properly understand concepts (patient capacities or states) and being able to act accordingly (behaviors) [1]. This conceptual model of empowerment by Bravo et al. [1] includes “use the internet to collect/share health information and support”.

The term empowerment is widely used in the literature although there is no single definition, which makes consensus difficult. There are numerous questionnaires to assess a patient’s empowerment but there is no consensus on which one to use [1,3,4,5].

To achieve this empowerment there are several strategies among which the digital environment can be included as an excellent resource to access information of interest and promote autonomy; in this case we would speak about digital empowerment.

Digital empowerment helps people to increase their level of ehealth literacy which is defined by Chong et al. (2020) [6] as the person’s ability to search for and understand health information on the internet in order to make decisions about health [6]. Ehealth literacy can be considered a useful tool for health education, promoting healthy lifestyles, reducing complications, or improving the follow-up of patients with chronic diseases [7,8]. However, do nurses use strategies to measure the empowerment of their patients? Do they consider the Internet as an ally in this task? How do nurses use ehealth literacy to increase patient’s empowerment?

Nurses are well positioned in the health care system to empower clients’ abilities to understand and utilize health information for their own health [6]. Information and training in health, as well as monitoring through face-to-face (F2F) or digital consultations carried out by the nurse are acts of communication and, therefore, it is key to guarantee that patients are empowered enough to handle such new situations.

There is currently a significant number of people who enquire into health issues through the Internet and social networks or who use the network to comment on issues related to their health [9], both for themselves and for their family or friends [10,11]. This phenomenon is of great interest and has been widely studied in recent years, especially because their internet consultations can significantly influence the decisions they make about care and self-care [9]. In fact, many people consult the internet when making serious health decisions, such as whether or not to undergo a surgical intervention [12], vaccinate their children [13], select a treatment for a family member in intensive care [14] or even make decisions as delicate as donating an organ from a living donor [15].

However, the internet is not only used to search for information; it is also frequently used to contact people who share the same interests or health problems, especially in relation to chronic diseases or cancer, and to form mutual help groups [16]. The exchanges among patients in online communities favors learning about the management of the disease, improves empowerment [17], helps to establish social support links and can even enhance adherence to treatments [18].

The use of the internet or digital tools not only helps chronic patients to improve their health when they share their concerns and doubts in online communities, but also seems to be associated with beneficial effects in individual interventions, improving adherence and self-management [19]. Nurses, as an active part of health communication with the patient, can suggest and recommend these spaces for exchange among patients to guide them, thus helping to increase their empowerment and health literacy.

The most frequently consulted social networks used by all age groups to retrieve health information are Facebook and YouTube according to some authors [9], although more recent studies also highlight the strong presence of WhatsApp as a resource for health information search and retrieval [10].

The topics they consult are very diverse, although they are mainly focused on information on different stages of life such as pregnancy or menopause, the management of chronic diseases such as diabetes or hypertension, or the use of medication [20], specific diagnoses or health problems, and to a minor extent a healthy lifestyle (exercise, nutritional recommendations, etc.) [9].

However, it is important to note that the fact that patients have adequate digital skills and that they consult health information on the internet does not imply that it is correctly understood [21] and even that can give erroneous results that may confuse those who use it [22]. Many users view health content on YouTube, although this content may be erroneous or confusing and may not provide quality information [23,24]. The problem is worsened when this information is not confirmed with a nurse or doctor, or when they do not show any interest in the information obtained by the patient [25]. As it can be frustrating for the patient to find contradictory information and not be able to talk about it, it is essential that the nurse be able to guide the patient in the search for quality information in order to reduce anxiety and minimize inadequate content consultations [26,27]. Contrary to what one might think, the professional–patient relationship can be strengthened if this confirmation on the Internet is used to speak openly during the consultation [28]. The main barrier pointed out by nurses when it comes to consulting and confirming valid information on the internet is a lack of time, followed by a lack of skills in this area [29,30].

However, one of the health care professionals’ main objectives should be to empower patients, offering them the best health-related information, for which it is essential to have a good understanding of the available resources and the proper tools to manage these. The use of technology in the health environment is now widespread, with thousands of resources, tools, and apps. The Internet gives nurses and doctors multiple possibilities to create their own resources for the patients’ empowerment. In fact, most healthcare professionals use the Internet as their main tool for acquiring new competences, learning, and sharing information [31].

In most countries, nurses are responsible for patients’ health education, especially for those with chronic diseases [32,33,34,35]. That implies monitoring to check that patients acquire the skills they need for self-care and self-management of their health, and verifying that they know and understand possible problems and how to apply solutions, etc. [36]. Nevertheless, few digital resources have been offered or created by nurses for patients, despite the fact that they are effective methods of monitoring the chronically ill [37]. This can be a privileged setting to better understand the needs and interests of patients or those at risk of becoming ill, can be used to understand their behavior in relation to health [38,39] and direct this knowledge to training the population and offering quality health advice [40]. We should not forget that the best way to develop quality resources is in partnership with the patient as an advisor and guide [41].

The aim of this exploratory study was to identify the strategies nurses use to digitally empower their patients, the tools, and resources they usually recommend, and the segments of the population most likely to benefit from these new channels of information and learning.

The initial hypothesis based on our own experience in the field is that nurses do not use any tool to estimate empowerment and that they rarely use digital tools with patients, especially elderly patients, as there is not a clear patient empowerment strategy in most of the healthcare service providers.

## 2. Materials and Methods

A survey was designed in several development and verification phases and was subjected to analysis by a panel of experts. It was then tested on a pilot group to analyze its performance and select the final items before the survey was carried out by digital means.

### 2.1. Initial Definition of the Survey Items by a Panel of Experts

To analyze the current situation of knowledge, competencies, and skills in the use of digital tools among health professionals, an initial questionnaire with 26 queries, mostly open-ended, was submitted for review, analysis and discussion by a panel of experts composed of thirteen expert researchers, disseminators and professionals directly related to this field of study. After receiving their feedback, the questionnaire was reduced to 24 questions, five of these related to demographic data (working place, gender, etc.). Open free answers from the experts were used to select the most common terms in order to adapt the questionnaire to a predefined set of closed answers and in this way the authors defined the answer format to obtain easily measurable and quantifiable results.

### 2.2. Testing the Survey with a Group of Health Professionals

This questionnaire, prepared on Google Forms was validated by the Research Ethics Committee of the Universitat Politécnica de València, Spain (P4_25_07_18), which simplified the storage of the answers and subsequent analysis.

The questionnaire was sent by email to the members of two medical associations of nurses and doctors from the Valencia region from December 2018 to February 2019 after obtaining their authorization. Sampling was thus obtained in a randomized way, without being aware of the previous digital health literacy level of the participants.

In the first step, 103 responses were obtained that were analyzed to detect possible deficiencies and redundant or complex questions.

Finally, it was decided to withdraw three of the items, which did not provide relevant information, leaving the final survey with 21 questions.

### 2.3. Structure and Details of the Survey

As in the first step, the questionnaire was hosted on Google Forms and therefore could be answered online from any device. The answers in this case were automatically stored for further processing. For data protection reasons it was decided not to collect the participants’ email addresses.

On the first screen, the participants accessed a brief explanation of the survey’s objective, the approval of the ethics committee and the data of the responsible persons. They also had to press the “I accept” button as consent before starting the survey itself. The estimated time to answer the survey was less than 5 min.

The definitive questionnaire was structured as follows: the first three items were related to demographic data, the next nine to Internet use and digital tools in the healthcare environment. Five questions dealt with the definition and measurement of the patients’ empowerment, while the last four items were based on the opinion of the healthcare professional of possible improvements in the healthcare system both with and without the Internet.

These items were grouped into four sections in which the participants could change their answers if they so wished before finally submitting them. The questions were mandatory and included several options to choose from, although they also had an ‘Others’ section to add additional comments.

### 2.4. Launch of the Large-Scale Survey

To calculate the necessary sample size, data were requested from the General Nursing Council of Spain, which estimated that there were 316,094 nurses in Spain in 2019. Starting from this population and at a confidence level of 99%, with a ± 5% margin of error (confidence interval), 665 surveys were required.

To achieve this objective, it was decided to launch an open survey focused on nurses from all age segments through social networks and other digital media posted on different networks such as Instagram, Twitter, Facebook, and LinkedIn. To reduce the possible bias if those who responded were only nurses present in social networks, and therefore with greater digital dexterity, it was also sent through popular mobile messaging systems. The only requirement for answering the survey was to be an actively employed nurse. Participation was completely voluntary, anonymous, and disinterested.

The survey was publicly released in December 2020 and was open for 13 days. We decided to close it after getting 850 responses.

Appendix A Table A1 includes the Checklist for Reporting Results of Internet E-Surveys (CHERRIES) [42] in which detailed information on the process carried out is described in detail.

### 2.5. Statistical Analysis

Different test statistics were applied to check if the results were statistically significant (*p* < 0.05). Most variables were categorical and some dichotomous. We used the chi-square as the test statistic with the corresponding degrees of freedom depending on the dimensions of the contingency table. Some variables, such as age, were considered ordered variables, grouped in different age intervals. In those cases, we used the Kruskall-Wallis test and correlation. Some questions in the survey were “check all that apply’’ (respondents select as many of the response options as are perceived to apply to them) for which we considered each answer as a dichotomous (Yes/No) categorical variable

## 3. Results

A survey was carried out to determine the strategies used by the nurses to empower their patients during the month of December 2020. This survey was launched openly and massively on social networks and instant messaging and 850 responses were obtained in 13 days. Two of the surveys were eliminated for inconsistent responses and the rest were 100% complete.

84% of the respondents were women (16% men), exactly the same proportion as in the population (nurses affiliated to the Spanish professional nursery association). 63% of the people who answered the survey were between 20 and 40 years old and 37% were over 40. The approximate proportions of the same two age groups in the population are 58% and 42%, respectively (the value is approximate since the association’s age groups are not exactly the same as in our survey, e.g., they have groups under 35, 35–45 and so on, while we used 20–30, 30–40 and so on. We assigned half of the people in this interval to each age group, i.e., we hypothesized a uniform age distribution for that interval). These values show the agreement between the real demographics in the population and the sample we used.

Additional metadata of the sample included the participants’ main professional activity: 60% of the respondents worked in a hospital environment, 25.4% in primary care, 7% in nursing homes and the rest in other places such as mental health clinics, occupational health, etc. As mentioned before, in order to avoid any bias in the results due to the online nature of the survey, we used different channels to recruit the participants, so the survey was not completed only by people with high internet/social network activity. We controlled this issue by explicitly asking the participants about their technological level. 14% scored themselves as only regular in technology, which meant that not only digital experts answered the survey.

### 3.1. Definition of Empowerment: Which Adjectives Do Nurses Associate with an Empowered Patient?

The first question addressed the problem of how nurses understand the empowerment concept. After a study of the state of the art of empowerment [1,3,4,5] and a previous testing survey with more than 100 key stakeholders, we validated that the concept of empowerment turns around four adjectives, as also identified by Barr [3] and Bravo [1]: informed, active, responsible and autonomous. We also included the option that all of these apply, and for those not identified with the aforementioned adjectives we offered the possibility of writing an open text response. The proportion of people that marked the corresponding response was: informed (19.81%), active (15.8%), responsible (10.02%), autonomous (15.68%), and all of these (38.68%) (see Figure 1).

It is important to remember that the response to this question was not limited to the list of aforementioned adjectives; they were also allowed to write in open text format other descriptors for the empowerment idea. Only one did not mark any of the proposed adjectives, and two used closed terms to the proposed ones (they used the word ‘proactive’ instead of ‘active’). The fact that this open text option was not used by the participants showed that the first survey to refine the questionnaire was successful, since the list of proposed adjectives to describe empowerment included almost everybody’s answer. In fact, since the most frequent answer (almost 40%) was “all of these”, we can see them as a kind of thesaurus of empowerment.

### 3.2. Empowerment Measurement: Which Method Do Nurses Use to Assess Empowerment?

The next question aimed to discover the tools that nurses use to measure empowerment in patients. This was a check-all-that apply question with five options. Three were based on previous knowledge of how empowerment is usually measured: educational level, attitude to new information and/or challenges, and the ability of the patient to make decisions. Another possible answer was an open text format so that the participants could explain the way they do this. And the last possible answer was “none” for those that did not measure patient empowerment.

The participants selected all the answers that applied. Only approximately one out of ten (9.19%) did not use any tools to measure empowerment (they checked the “none” option). The three proposed measures covered almost all of the possible methods that nurses use to measure empowerment (only 3.53% chose the open text answer, i.e., use additional tools to the proposed methods). The most common ways of measuring empowerment were attitude (61.9%) and ability to make decisions (71.8%). Educational level was also used, but to a lesser extent (18.2%).

We wished to determine any association between the way the nurses described the empowerment concept (responses to the previous question) and the way they measured it (this question). We did not find any statistically significant differences for the people that did not measure empowerment (*p* = 0.630) or for those using the attitude to new information and/or challenges (*p* = 0.632). However, we did find differences for those using the ability of the patient to make decisions (*p* = 0.003) and an even stronger association for those measuring empowerment based on the educational level (*p* = 0.00024).

For the nurses using the patient’s ability to make decisions, the most descriptive adjective of the empowerment idea was ‘autonomous’, while those relating educational level to empowerment used all the adjectives to describe empowerment. Table 1 shows the exact percentages for the different options for these two statistically significant groups. In other words, those using the educational level did not discriminate between the adjectives (they have a general idea of empowerment, 25% checked the all option), while those in the group that used the ability of the patient to act considered that, although all adjectives applied (the minimum value was very high, 65% for informed), the best defining term was the autonomous character of an empowered patient (as expected, considering that they identified empowerment mostly with patients who make their own decisions).

More detailed responses to this section can be found in Appendix A (Table A2a,b).

### 3.3. Profiling the Patients: Which Patients Ask for More Information?

Once the definition of empowerment and the way the nurses perceived it was established by the previous questions, we needed to go deeper in order to obtain a profile of the empowered patient. This was measured by the question: what kind of patients ask for more information? This aimed to find any differences due to the sex, age and health status of the patient.

With respect to the sex of the empowered patient, the participants considered that women demand more information than men (41% vs. 13%). Since the percentage of women nurses is much higher than men (84% vs. 16% in the sample and in the population), we needed to check if the sex of the participants was a confounding factor. 48 of the surveyed men (35.82%) and 303 of the women surveyed (42.44%) said that women demanded more information than men. Although it is true that female nurses are more likely to consider that women patients are more curious, there was no statistically significant difference (*p* = 0.153), showing that the sex of the participant in the survey did not alter the main finding that female patients are more likely to ask for more information.

We also studied the dependency with the patient’s age. In this case we used two groups, younger and older than 60 years of age. The elderly people asked for less information (13.91% vs. 42.80%). In this case, we could consider that age influenced these results, since more young nurses answered the survey. As we did with sex in the previous paragraph, we checked that the age of the participant in the survey had no statistically significant influence on the result. In this case, the percentages of the respondents that considered that younger patients demanded more additional information were: 45.83% (respondents’ age between 20 and 30 years old), 40.54% (between 30 and 40), 46.70% (between 40 and 50), 35.08% (between 50 and 60) and 27.77% (over 60).

Another variable that can help to profile empowered patient is his health status. We asked the nurses if they considered that patients with a chronic pathology were more challenging and 42.57% of the respondents agreed that this type of patient asks for more information than the others.

### 3.4. Frequency: How Often Do Patients Ask for More Information?

After profiling the empowered patient, we needed to know the level of the patient’s activity; i.e., how often they asked for more information. In this question we used a Likert frequency scale with four options: always, usually, rarely, and never. The results are shown in Table 2. The most relevant answer was that 63.44% considered that patients rarely ask for more information; i.e., that when empowerment is associated with an active, informed and autonomous patient, it is not translated into an extended use of these features.

### 3.5. Communication Channels: How Do Nurses Provide the Extra Info Demanded by the Patient?

Once we knew that only a small percentage of patients asked for extra info, we needed to evaluate whether the way that the extra information was provided to the patient had an influence on this fact. We asked the nurses which method they used to provide the additional information; e.g., we could hypothesize that if the channel used to give the information was too complicated or difficult to understand, this could explain the low percentage of those who asked for more information). Since there were different methods, the respondents could mark several options (all that applied to the question). The proposed methods were printed material, oral communication, multimedia content (video and images), webpages (links), apps (mobile), and others. Understanding the characteristics of these from different points of view, such as complexity, required technological skills, durability, reproducibility, etc. We assumed that all of these were trusted sources, since they were provided by nurses (we were not interested in measuring the relevance of the app or webpage; we assumed that they were previously filtered by the professional). The results are in shown Table 3, where it can be seen that printed documents were the most common method (71.93%), followed at a great distance by oral communication (40.09%); i.e., the traditional methods (written and oral) were preferred to technological channels; for example, the use of printed information (71.93%) was double the use of multimedia (30.18%) or Internet (29.95) content.

The results given in Table 3 showed that the current digital era does not translate into the use of more technological tools when providing information in medical environments. It was therefore important to check whether there was an association between the method used to give the information and the age of the nurse, i.e., if this resistance to introduce more modern channels occurred in all age groups or if it was a generational issue that could be fixed by time in a natural way and did not require institutional intervention. Table 4 shows the percentages (%) of different media for each age group of the respondents.

The results for each medium were analyzed to find any differences between the different age groups using the age groups as an ordinal variable (mean values 25, 35, 45, 55, 65) and the Chi squared test using only two simple intervals: under and over 40 years of age, considering 40 as a simplified young/old threshold.

Contrary to the belief that young nurses are more familiar with a digital way of living and would use the same digital approach in their professional life, we found that printed material was used equally by nurses in all the age groups, with no difference (*p* = 0.85, Mann–Whitney U test). In fact, when grouped in under and above 40, the proportions were quite similar: 70.60% and 74.20%, respectively, with no statistically significant difference (*p* = 0.259, Chi squared). A similar result was obtained as regards providing verbal information (38.22% under 40 and 41.20% above 40 use oral communication (*p* = 0.39)), showing that the traditional channels were used in the same proportions by all age groups.

With respect to multimedia content, although not statistically significant (*p* = 0.082), there was a small difference: 28.09% (under 40) vs. 33.76% (above 40), i.e., older nurses used more multimedia content. In fact, this tendency was amplified when the information channel was more technological. Figure 2 shows that the median age of nurses that did not provide Internet links was 35 years old, while those that recommended websites was 45, as we expected from the growing percentage with age in Table 4 for web pages.

In the younger and older than 40 groups, the percentages were 23.60% and 40.76% (*p* < 0.001), respectively. Nurses under 40 were less than half as likely to recommend a website to obtain more information than the older nurses (OR 0.45, 95% CI 0.33–0.60).

This behavior was repeated for other technological teaching tools. For example, only 17.31% of the nurses between 20 and 30 years recommended mobile phone apps to provide access to more information requested by the patients (see Apps column in Table 4 to confirm the increasing use of this tool by the group between 51 and 60, with statistically significant differences between the age groups (Mann–Whitney U test *p* = 0.001). As expected, when grouping into only two categories using 40 years as the threshold between the classes, the difference in the percentages and odds ratio are statistically significant (18.16% under 40 vs. 27.39% above 40, *p* = 0.002, OR 0.59, 95% CI 0.42–0.81).

Based on the results we could affirm that our initial hypothesis, “The nurses do not use any tool to estimate empowerment and that they rarely use digital tools with patients, especially elderly patients” is initially fulfilled even though longer time and larger size studies are desirable to bring stronger evidence. On one hand, we observed that most of the nurses did not apply validated questionnaires for the estimation of empowerment, although it is interesting to note that they use other methods such as observation and knowledge of the patient.

On the other hand, we observed that nurses still use few digital resources (See Table 3). This is likely to be related to the age of the patient, as we expected, but also to the age of the nurse. Based on this result, a new variable appears that should be taken into account for future research. 

## 4. Discussion

### 4.1. Nurses’ Assessment of the Patient Empowerment Level

The concept of empowerment continues to be ambiguous and controversial for health professionals and the numerous ways of defining this multidimensional construct have been analyzed [1,3,4,5]; however, most of the published definitions include aspects such as those selected in this study: i.e., informed, active, autonomous and responsible. It is important to bear in mind that the individual option most often chosen by the nurses participating in this survey was “informed”, which not only implies action and responsibility on the part of the patient, but also directly affects the professional–patient relationship. “Healthcare providers have responsibilities to respect patient autonomy and adopt a partnership style within the healthcare relationship [1]”.

Stating clearly that there is not a single accepted definition of patient empowerment, it was highly valuable to analyze the strategies the nurses have to measure the level of empowerment of their patients, where most of the strategies were subjective and were based on the perception from the nurse of the patient’s ability to make decisions. Only 3% used other types of tool, such as questionnaires or surveys. This could be due again to the ambiguity of the concept, the great variability of the existing instruments for its measurement [1,3,4] and the scarcity of validated questionnaires in the Spanish language in this specific case [43]. According to the survey, almost 10% of Spanish nurses do not use any type of parameter to measure their patients’ level of empowerment.

However, as Garcimartín [43] has pointed out, due to their competency related features, nurses are the best prepared to promote and facilitate support in order to accompany patients in their selfcare and empowerment process.

### 4.2. Empowered Patient Profile and Demand for Information

According to the Spanish nurses surveyed, average profile is women under 40 years of age with a medium-high educational level and a chronic ailment among those who most request additional information. These data agree with the results of Bidmon [44], who analyzed the request and search for health information through the Internet in the German population and found that women were the most interested in this information. In a survey of 18,497 people, Wynn et al. [45] also found that women around 40 years of age with a pathology were the largest group who searched for medical information on the internet.

### 4.3. Perception of eHealth Literacy of Older People in Younger Health Nurses

This survey carried out on Spanish nurses during the COVID epidemic showed that digital sources of extra information offered to patients, such as videos, links, or applications, contrary to general belief, are mostly suggested by nurses in the older age groups. This could indicate a shortfall in the training of younger nurses or perhaps a generation gap [46]. Studies such as that by Kim et al. [47] suggest that those with little medical knowledge in fact find mobile applications acceptable and easy to use for educational material, detection questionnaires and measuring literacy.

This has also been pointed out by Heiney et al. [48], who concluded that semi-literate patients with reduced medical knowledge and limited experience of smartphones can benefit from the use of mobile applications to improve their self-care, even those older than 60 years.

To address this misperception of ehealth literacy of older people in young nurses, different actions should be proposed, especially in the final years of the degree course or in the initial years of their health care practice

### 4.4. Strategies to Increase Patients’ eHealth Literacy

According to our survey, in order to increase patients’ ehealth literacy (considering all the age ranges, not only older people), the methods used mostly by nurses to offer additional information to patients after the consultation or clinical intervention are still printed information on paper and oral explanations. However, there is much evidence that supports the usefulness of elements such as videos or telecare [49], mobile applications [50,51,52] and web-page recommendations to reinforce the message offered in face-to-face meetings. These types of tool are very useful in promoting self-management in patients, especially with chronic diseases. Therefore, this evidence needs to be more visible and accessible in order to persuade nurses to fit such empowerment strategies and use more multimedia material and not only printed.

## 5. Limitations

This study was conducted during the emergency pandemic situation. This may have influenced the lack of measurement of nurse empowerment. However, the pilot study to test the survey was conducted in 2019 (pre-pandemic) and the results in terms of people not using any method to estimate the empowerment of their patients was higher (17.5% vs. 9.3% of the 2020 survey).

One inherent limitation is that we cannot check the effectiveness of this precautionary measures to avoid any bias in the sample answering the survey. We expect that, due to the large of the sample size, any bias has been minimized.

## 6. Conclusions

Although numerous studies point out the advantages and benefits of introducing digital tools as a complement to the health education offered by nurses and doctors in consultations, these methods are still not widespread among health professionals.

Even though measuring patients’ empowerment can be useful when assessing their understanding and capacity for self-management, the majority of the Spanish nurses surveyed do not use any standardized method for this. It would be interesting to carry out similar studies in other countries in order to make comparisons.

Finally, it is interesting to note that nurses do not use standardized tools to measure empowerment but they do use patient observation and analysis. It could be interesting to design useful and usable tools based on the real needs of nurses.

According to our study, younger professionals consider that the patient’s age and scarce knowledge are the major impediments to applying digital methods to reinforce the health message and promote self-care. Older nurses recommend more mobile apps or websites to their patients than their younger colleagues.

It would thus be advisable to train health professionals in measuring their patients’ empowerment and in preparing digital health materials, as well as working on the prejudices and preconceptions of younger nurses with respect to older patients, since they are the ones who can benefit most from continuous monitoring and follow-ups.

Some studies mention similar findings, such as that of Van Houwelingen [46], but more studies are needed to check whether the age of the nurse is really a determining factor when applying digital resources to educate the patient.

## Figures and Tables

**Figure 1 ijerph-18-09844-f001:**
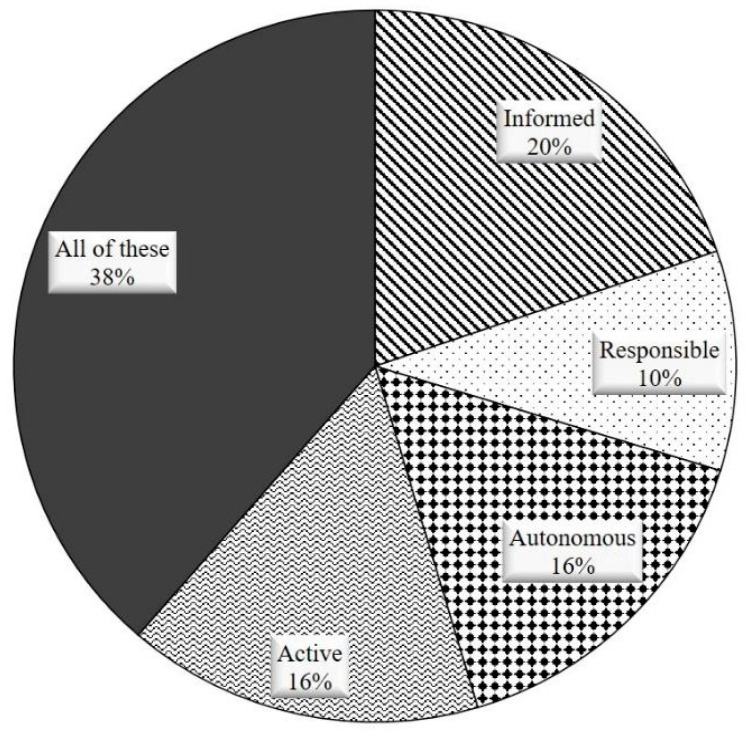
Empowerment definition percentages.

**Figure 2 ijerph-18-09844-f002:**
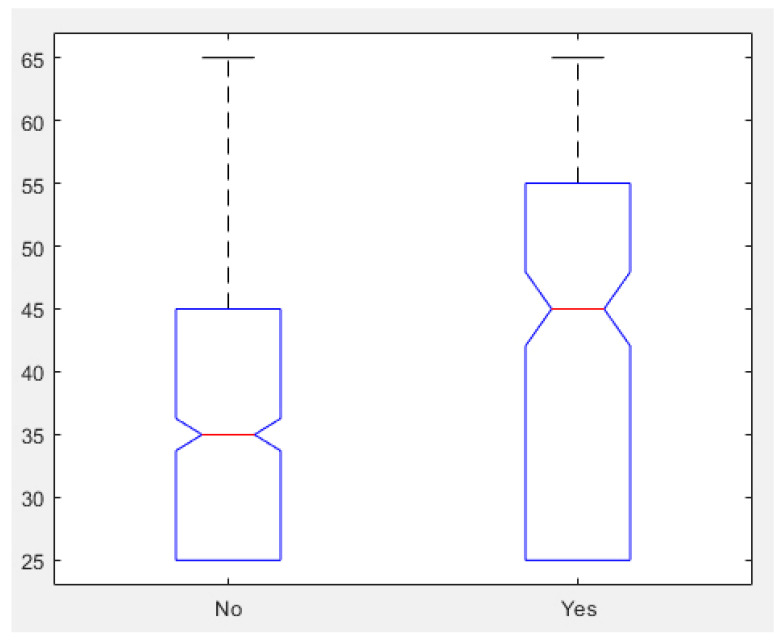
Median age of nurses that use (Yes) and do not use (NO) websites.

**Table 1 ijerph-18-09844-t001:** Description of empowerment for nurses using the ability to make decisions and educational level as measures of empowerment (%).

	Active	Autonomous	Informed	Responsible	All of These
Ability to take decisions	66	82	65	66	75
Educational level	11	12	14	18	25

**Table 2 ijerph-18-09844-t002:** Frequency of asking for more information.

Frequency	Number of Respondents	%
Never	9	1.06
Rarely	538	63.44
Usually	295	34.78
Always	6	0.70

**Table 3 ijerph-18-09844-t003:** Communication channel used to provide additional information.

Communication Channel	Used by	%
Printed material	610	71.93
Oral communication	340	40.09
Video and images	256	30.18
Web pages	254	29.95
Apps	183	21.58
Others	47	5.54

**Table 4 ijerph-18-09844-t004:** Percentage of communication channels for different age groups.

Channel	Printed Material	Oral Communication	Video and Images	Web Pages	Apps
20–30	74.78	40.06	30.77	21.79	17.31
31–40	64.86	42.79	24.32	26.13	19.37
41–50	70.88	36.26	31.32	34.62	24.18
51–60	78.95	44.74	38.60	48.25	33.33
+60	77.78	16.67	27.78	55.56	22.22

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
