# Peer review of "Estimating Patient Empowerment and Nurses’ Use of Digital Strategies: eSurvey Study"

_ijerph, 2021, doi:10.3390/ijerph18189844_

Round 1
Reviewer 1 Report
The article is interesting and promising, but there are some problems of coherence in the approach that need to be resolved:
*The title and abstract are not directly related to the content of the research: the article does not study either the prejudices or the conditioning factors of nurses' ICT use.
*The analysis of the results is articulated around the concept of patient empowerment. This concept is not developed operationally in the theoretical framework or in the introduction, so we do not know why it is important. The concept needs to be critically defined and its importance in relation to nurses' use of ICTs should be justified.
*I think it is complicated to talk about nurses' prejudices, from a theoretical perspective. There is no variable as such in the research, so the researchers' biases in analysing the data (not the data itself) speak about nurses' prejudices (which is a big mistake from a research perspective).
*The conclusions do not involve either a discussion with the knowledge so far on the subject, nor do they provide proposals for implementation. They are only a superficial summary that does not contribute knowledge.
Reviewer 2 Report
Dear Authors!
Thank you for taking up a very interesting topic of „Analysis of nurses’ prejudices and conditioning factors for the use of digital tools”. Some elements of the presented study are valuable (e.g. large sample size) but overall the text requires extensive corrections.
The list of issues to be improved is presented below:
- Such key terms as "health literacy"; "digital literacy" or "empowerment" should be derived from the literature on the subject and precisely defined. This is missing from the text. The same applies to empowerment measurement. The authors state that they conducted a literature review, the result of which let to indicate "four adjectives usually associated with the empowerment idea" / lines 218-219 /. It is worth emphasizing it in the theoretical part concerning the concept of empowerment.
- It is a pity that the authors, after analyzing the literature, did not make any preliminary hypotheses.
- The aim of the article is not entirely clear and correctly indicated: "The aim of this study was to determine the way in which nurses measure their patients' level of empowerment, the use they make of digital tools to reinforce their messages and whether this varies according to the sex or age of both patients and nurses ". I have considerable doubts that in the harsh conditions of the pandemic, nurses take time to "measure their patients' level of empowerment". We cannot say that nurses measure their patients' level of empowerment, and rather it is about estimating it.
- The title of the article does not fully reflect its content.
- Is this statement supported by some research?: „In most countries, nurses are responsible for patients’ health education, especially for those ones with chronic diseases” /lines 105-106/
- What are the limitations of the study performed? (e.g. non-random sample selection which affects limited representativity, although the sample size was impressive).
- Appendix A is redundant, and it does not bring much new to the information contained in the text. Instead, it is better to add a table with detailed responses in the empowerment measurement section (section 3.2.)
As a result, an article on a larger scale bears the hallmarks of a research report than a research paper. I encourage authors to improve the text.
Best regards,
The reviewer.
Round 2
Reviewer 1 Report
Dear editor and authors,
Thanks for giving me the opportunity of reviewing this second version. My opinion is that authors have reached high level improvements, and with them they have responded every single question I had. My main concerns about some misunderstandings, prejudices and assumptions are missing now, so I consider the paper can be published since it offers a step further in the creation of scientific knowledge.
Well done!
Author Response
Dear reviewer,
we would like to thank you for your considerations that have allowed us to improve the article and learn more about how to develop our future research.
Thank you very much
Reviewer 2 Report
Dear Authors!
Thanks for considering my remarks. I find the presented changes and explanations appropriate.
In my opinion, the paper in its current form presents a much higher scientific quality, although the empirical data was subjected only to basic statistical analyzes, not using statistical tests.
Before the final acceptance of the paper, I recommend only one minor change, but it is necessary. In empirical research, when we put preliminary hypotheses, it is necessary to refer to them clearly in the results section or conclusions section. Based on the research results obtained, it is worth indicating whether the hypothesis was confirmed or maybe rejected. As a result, it will translate into paper's readability. I suggest adding such a sentence, where you refer to the hypothesis set prior to conducted research (was it confirmed or not).
Best regards,
The reviewer.
